# Breast Self-Examination Practice and Its Determinants among Women in Indonesia: A Systematic Review, Meta-Analysis, and Meta-Regression

**DOI:** 10.3390/diagnostics13152577

**Published:** 2023-08-02

**Authors:** Yohana Azhar, Ricarhdo Valentino Hanafi, Bony Wiem Lestari, Freda Susana Halim

**Affiliations:** 1Faculty of Medicine, Universitas Padjadjaran, Bandung 40161, West Java, Indonesia; freda20001@mail.unpad.ac.id; 2Department of Surgery, Oncology, Head and Neck Division, Hasan Sadikin Hospital, Bandung 40161, West Java, Indonesia; 3Department of General Surgery, Hasan Sadikin General Hospital, Bandung 40161, West Java, Indonesia; ricarhdo22001@mail.unpad.ac.id; 4Department of Public Health, Faculty of Medicine, Universitas Padjadjaran, Bandung 40161, West Java, Indonesia; bony.wiem@unpad.ac.id; 5Department of Surgery, Faculty of Medicine, Pelita Harapan University, Tangerang 15811, Banten, Indonesia

**Keywords:** breast cancer, breast self-examination, prevalence, Indonesia, meta-analysis

## Abstract

Breast cancer (BC) is a heavy burden for Indonesian healthcare, but there is still no thorough evaluation for Breast self-examination (BSE) practice as routine BC screening. In this study, we aimed to synthesize the pooled prevalence data of BSE practice, compare BSE practice prevalence between Java Island and non-Java Islands in Indonesia, and identify the determinants that we thought could affect the BSE practice in the Indonesian population. Intensive searches were conducted in Cochrane Library, PubMed, Google Scholar, and SINTA (Indonesian Web of Science and Technology Index) from September 2017–2022. We utilized Review Manager 5.4 for conducting the meta-analysis. We found the overall national prevalence of BSE practice was 43.14% (95% CI: 36.08, 50.20, *p* < 0.00001). BSE practice in Java Island was higher compared to non-Java Island (44.58% vs. 41.62%). The highest prevalence of BSE practice was found among university students, with a 49.90% prevalence. Good knowledge, good attitude toward BSE, family history of BC, family support, and BC information exposure were all statistically associated with a higher determinant of BSE practice. We concluded that BSE practice in Indonesia is still low, especially in non-Java Islands. Integrative and collaborative programs should be established to promote BSE as routine screening for BC.

## 1. Introduction

In Indonesia, BC has been a prodigious healthcare problem, as it is the most diagnosed cancer according to the World Health Organization (WHO) in 2020 [1,2]. Compared to other Southeast Asian countries, Indonesia was recorded for having the most newly diagnosed advanced stadium BC, affecting a poor prognosis and aggravating the healthcare burden [3,4,5]. These long-term issues were induced via low awareness of BC screening and education regarding the importance of mammography as an initial BC screening [2].

Breast self-examination (BSE), or *Periksa Payudara Sendiri* (SADARI) in Indonesian, is a once-a-month self-examination to inspect and evaluate any abnormalities in the breast [6,7]. The practice was initiated in the 1950s before the advanced tool mammography was introduced [6]. This examination is a simple, self-applied, at-home procedure to assist every woman with an early screening of their breasts to detect any differences, such as lumps [6,7]. Thus, early recognition and intervention could be achieved to avoid further complications at the earliest stage [6,7,8].

Although WHO and the Centers for Disease Control and Prevention (CDC) do not recommend this method as a regular screening, low-to-middle countries still encourage BSE practice as an early screening due to practicality, limited access, and unequal distribution of mammography [9,10,11]. Regarding mammography, unfortunately, the Indonesian Healthcare and Social Security Agency itself limits the standard BC screening such as mammography due to budget limitations [12,13].

Although BSE does not provide early detection for tumors in situ or <1 cm mass, it is still a practical diagnostic method to find a palpable mass, and therefore, it is still used and encouraged widely in Indonesia. Therefore, the authors thought it is necessary to explore a thorough evaluation of BSE practice prevalence in Indonesia, with the fundamental purpose of providing a proper BC screening and extenuating health care burden.

To understand the background of this research, one should also understand the uniqueness and complexity of healthcare in Indonesia. A population of 270 million people (2022) makes Indonesia the 4th most populous country in the world, with 60% of them living on Java Island [12]. Java Island is the most habituated island in the country, and its geographical nature is relatively simpler than other bigger and more remote islands, such as Kalimantan and Papua [12]. Economic centralization in Java Island worsens the issue, and the accessibility to proper healthcare facilities remains inequitable [12,14].

The country also consists of more than 1000 ethnic groups living on 17,000 islands; even within one island, many ethnic groups live together in harmony, although they have different cultural beliefs [12]. Such cultural differences in Indonesian society also contribute to the healthcare gap, as not every region on the same island, for instance, has the same opinion toward healthcare and awareness. All of the aforementioned factors contribute to the overall health issue awareness on non-Java Islands staying flattened and, therefore, need to be evaluated [12,13,14,15,16].

With such backgrounds and with the spirit to give better breast cancer screening to all Indonesian population, this study aims to synthesize the pooled prevalence data of BSE practice, compare BSE practice prevalence between Java Island and non-Java Islands in Indonesia, and identify the determinants which we thought could affect the BSE practice in Indonesian population.

## 2. Materials and Methods

### 2.1. Eligibility Criteria

For a systematic review of prevalence, we generated a CoCoPop (condition, context, population) framework to define the inclusion criteria [17]. (1) Condition: Breast self-examination (BSE); (2) Context: Java Island vs. non-Java Islands in Indonesia within the last five years of studies; (3) Population: Any eligible Indonesian women. Furthermore, we included observational studies (cross-sectional, cohort, and case controls) with English and Indonesian language restrictions.

We defined the exclusion criteria as follows: (1) unpublished articles, (2) studies with no outcome variable, (3) studies without mentioning the study population and qualitative studies, and (4) articles that were evaluated as low-quality studies.

This study followed the Preferred Reporting Items for Systematic Review and Meta-Analysis (PRISMA) [18] and registered in the Prospective Register of Systematic Reviews (PROSPERO) with registration number CRD42022362907.

### 2.2. Information Sources

We conducted intensive searches in Cochrane Library, PubMed, Google Scholar, and SINTA (Science and Technology Index)—Indonesian web-based research information from September 2017–2022.

### 2.3. Search Strategy

The following keywords for conducting the searches are “breast self-examination”, “BSE”, “Periksa Payudara Sendiri”, “SADARI”, “prevalence”, and “prevalensi”.

### 2.4. Selection Process

Two authors (YA and FSH) performed the preliminary screening of titles and abstracts independently. Any disagreement was consulted and deliberated with the senior author (BWL) until a final agreement was reached.

### 2.5. Data Collection Process

The qualified studies were assessed for full-text screening. Any study that matched the inclusion criteria would undergo data extraction (authors’ details, year of publication, patient’s characteristics, study design, and outcomes measured) via YA and FSH. Afterward, BWL evaluated and determined the included studies for qualitative and quantitative synthesis.

### 2.6. Data Items

The primary outcome is the prevalence of BSE (the proportion of a population who examined their breasts to detect abnormalities). The secondary outcomes are independent variables related to BSE practice, including knowledge about breast cancer, attitude toward breast cancer, family history of breast cancer, family support on BSE practice, information exposure on BSE, and educational status.

### 2.7. Study Risk of Bias Assessment

We used the Joanna Briggs Institute (JBI) critical appraisal checklist for studies reporting prevalence data to assess the methodological quality of a study and to identify the possibility of bias in its design, conduct, and analysis [17,19]. YA and FSH critically appraised the included studies for qualitative synthesis. There are nine questions with four standard answers (yes/no/unclear/not applicable), and overall appraisal will be judged with three categories (high/moderate/low quality) via YA and FSH. Total points of 1–3 are considered low quality, 4–6 points as moderate quality, and 7–9 points as high quality. Moreover, BWL will resolve any disparity.

### 2.8. Effect Measures

Prevalence data (primary outcome) are reported as a proportion, and the resultant of the meta-analysis will be categorized as prevalence with a 95% confidence interval (CI). The additional correlation of BSE practice with independent variables is described in odds ratio (OR) with a 95% CI. The summarized primary and secondary outcomes synthesis will be presented in forest plots.

### 2.9. Synthesis Methods

We utilized Review Manager 5.4 software for synthesizing the outcomes measured. A generic-inverse variance with random-effect models, assuming the individual study prevalence estimates follow a normal distribution to analyze the standard error (SE) and 95% CI. We calculated the SE from the prevalence using the following formula [20]:$$SE=\sqrt{\frac{p\left(1-p\right)}{n}}$$

*p* = *Prevalence*

*n* = *Sample size*

A Mantel–Haenszel (M–H) formula with random-effect models was generated to synthesize the odds ratio (OR) and 95% CI for dichotomous variables. The heterogeneity of the results was explored using I-squared (*I*^2^) statistic, and overall judgments are categorized as low (0–25%), moderate (26–50%, and high (51–100%) degrees of heterogeneity.

We conducted a meta-regression with a random-effects model using a restricted-maximum likelihood for pre-specified variables, including regions, study population, publication year, and sample size, to identify the interaction effect of these variables in influencing the prevalence of BSE practice.

### 2.10. Publication Bias

We employed a funnel plot to assess the existence of qualitative publication bias. Egger’s regression test and Begg’s rank correlation method evaluate the asymmetry of the funnel plot in quantitative measurement.

## 3. Results

### 3.1. Study Selection

A total of 6994 records were identified using several databases (Google Scholar, PubMed, SINTA, and Cochrane Library). Initially, we identified 577 articles and proceeded to the screening titles, abstracts, and duplication, which excluded 221 articles. Following the full-text review, 310 articles were excluded due to the unavailability of reports on the outcome of interest, and five studies were assessed as low quality of study. Lastly, the final 41 articles were eligible for qualitative and quantitative synthesis (meta-analysis). The summarized flow chart is presented in Figure 1.

### 3.2. Study Characteristics

The total sample size from 41 full-text publications is 6361 patients, which comprised 4414 and 1947 patients based on Java and Non-Java Islands, respectively [21,22,23,24,25,26,27,28,29,30,31,32,33,34,35,36,37,38,39,40,41,42,43,44,45,46,47,48,49,50,51,52,53,54,55,56,57,58,59,60,61]. The study setting differed into four groups, including university-based (11 studies), school-based (11 studies), public health center-based (seven studies), and community-based (12 studies). All the studies were classified as cross-sectional designs. The detailed data characteristics of the included studies are presented in Table 1.

### 3.3. Risk of Bias in Studies

We conducted the study quality assessment using the JBI critical appraisal checklist for studies reporting prevalence data. All included studies’ scores ranged from 7–9, classified as high-quality (Appendix A).

### 3.4. The Pooled Analysis of Breast Self-Examination Practice Prevalence in Indonesia

The overall national prevalence of BSE practice was 43.14 [95% CI: 36.08, 50.20], *p* < 0.00001, followed by high heterogeneity with *I*^2^ = 100% (Figure 2). The lowest BSE practice prevalence was 0.91 [95% CI: 0.89, 0.93], observed by Cane PS et al. in 2019 in Aceh (non-Java region) [27], and the highest percentage was 90.00 [95% CI: 89.86, 90.14] conducted by Kurniawati T et al. in Lampung (non-Java region) [42]. The heterogeneity of the pooled analysis was considered high with *I*^2^ = 100%.

We conducted a subgroup analysis to compare the prevalence of BSE practice between two regions (Java vs. Non-Java), study settings, and population. Consequently, Java Island showed a higher percentage of prevalence in the determination of BSE practice with 44.58 [95% CI: 35.42, 53.75], *p* < 0.00001 compared to Non-Java Islands 41.62 [95% CI: 29.83, 53.41], *p* < 0.00001 (Figure 3A,B). The highest and lowest prevalence of BSE practice in Java Island was 74.47 [95% CI: 74.35, 74.59] observed by Zulaika C et al. [61] (2021) and 8.17 [95% CI: 8.15, 8.19] in Puspitasari YD et al. [46] (2021), respectively. Moreover, the pooled synthesis prevalence of BSE practice on Non-Java Islands revealed the highest percentage was 90.00 [95% CI: 89.86, 90.14] observed in Kurniawati T et al. [42] (2021) and the lowest prevalence demonstrated by Cane PS et al. [27] (2019) at 0.91 [95% CI: 0.89, 0.93].

We differed the prevalence analysis of BSE practice based on the study population and settings into four groups, comprising (1) university students, (2) school-aged students, (3) fertile women and public-health-center-based (PHC), and (4) fertile women in communities (Figure 4A–D). The following results were 49.90 [95% CI: 40.27, 59.33], *p* < 0.00001 from university students as the highest practice, followed by school-aged students with 42.51 [95% CI: 32.38, 52.64], *p* < 0.00001, fertile women in PHC at 40.66 [95% CI: 20.76, 60.56], *p* < 0.00001, and the lowermost practice at 39.06 [95% CI: 25.61, 52.50], *p* < 0.00001 from fertile women in communities group.

### 3.5. The Determinant Factors of Breast Self-Examination Practice in Indonesia

#### 3.5.1. The Correlation between Breast Self-Examination Practice and Knowledge about Breast Cancer

Thirty studies (*n* = 3131) were assessed to generate a pooled analysis in which women with a good knowledge about breast cancer were statistically meant to practice BSE than those who had a poor knowledge [OR 4.82 (95% CI: 2.84, 8.18) *p* < 0.00001, *I*^2^ = 82%] (Figure 5A).

#### 3.5.2. The Correlation between Breast Self-Examination Practice and Family History of Breast Cancer

Eight studies (*n* = 2920) were assessed to generate a pooled analysis in which women with a family history of breast cancer were statistically meant to practice BSE than those who had no family history [OR 1.80 (95% CI: 1.15, 2.82) *p* = 0.01, *I*^2^ = 54%] (Figure 5C).

#### 3.5.3. The Correlation between Breast Self-Examination Practice and Family Support

Six studies (*n* = 833) were assessed to generate a pooled analysis in which women who had family support on BSE were statistically meant to practice BSE than those who had no family support [OR 4.90 (95% CI: 3.43, 7.00) *p* < 0.00001, *I*^2^ = 0%] (Figure 6A).

#### 3.5.4. The Correlation between Breast Self-Examination Practice and Breast Cancer Information Exposure

Nine studies (*n* = 1139) were assessed to generate a pooled analysis in which women who were being exposed to breast cancer information were statistically meant to practice BSE [OR 2.64 (95% CI: 1.08, 6.48) *p* = 0.03, *I*^2^ = 85%] (Figure 6B).

#### 3.5.5. The Correlation between Breast Self-Examination Practice and Educational Status

Five studies (*n* = 2533) were assessed to generate a pooled analysis in which women who had higher education status (vocational education or university) were statistically meant to practice BSE [OR 2.78 (95% CI: 1.70, 4.55) *p* < 0.0001, *I*^2^ = 54%] (Figure 6C) than those who had lower education status (up to senior high school).

### 3.6. Meta-Regression

We identified the risk factors influencing BSE practice’s prevalence with meta-regression. Our meta-regression revealed that variability in that outcome in the Indonesian population could not be explained by known study factors associated with predictors of prevalence rates (Appendix A). From our meta-regression analysis, it was revealed that the prevalence of breast self-examination in Indonesia was not significantly influenced by regions (*p* = 0.6597) (Appendix A), study population (*p* = 0.7407) (Appendix A), publication year (*p* = 0.1518) (Appendix A), nor sample size (*p* = 0.7352) (Appendix A).

### 3.7. Publication Bias

The national prevalence funnel plot demonstrated a symmetrical inverted plot distribution (Figure 7). Egger’s regression test (*p* = 0.0001) and Begg’s rank correlation method (*p* = 0.0246) revealed statistically significant results. These findings indicated no qualitative and quantitative publication bias within our meta-analysis on the overall national prevalence of BSE practice.

This section may be divided into subheadings. It should provide a concise and precise description of the experimental results, their interpretation, as well as the experimental conclusions that can be drawn.

## 4. Discussion

In this research, we hypothesized that the overall BSE practice prevalence would be inferior to other countries. Additionally, we also hypothesized the non-Java BSE practice prevalence is expected to be lower than in the Java Island population due to the ongoing educational inequality in Indonesia, which influenced the current behaviors and awareness towards health practice [62]. Based on our knowledge, this is the first systematic review and meta-analysis to analyze the prevalence of BSE practice in Indonesia and to compare the Java and non-Java Island populations.

Although policies from WHO, CDC, and developed countries do not support BSE practice as a primary screening tool—several Asian countries, including Indonesia, continue to acknowledge the importance of BSE practice since mammography is not accessible equally among regions [30,31,63,64]. A study by Thaineau et al. in Thailand tracked a 1.9 million population regarding the risk of late-stage breast cancer in five years (2012–2017). They proved that non-regular BSE practice patients had a 1.319-fold higher risk than regular BSE practice patients [65].

Our overall national BSE practice prevalence (43.14%) was significantly lesser than in several studies. Amegbedzi et al. in Ghana showed an overall prevalence of BSE practice in adult women was 65% [10]. Consistently, a cohort study from Thailand demonstrated a 72% prevalence of BSE practice in well-informed patients and a slightly higher percentage (49.43%) in the study conducted in rural India in 2022 [63]. Nevertheless, our findings were higher than in Vietnam (15.25%) and Bangladesh (21%) [11,66]. For the population in Europe, we found a small study by Woynarowska–Soldan et al. (2019) [67] conducted in Polandia that concluded regular BSEs were performed only by 56.7% of the nurses. Almost half of the nurses failed to perform BSE due to unavailable information during the nursing profession [67].

Consistent with our hypothesis, we observed a higher prevalence of BSE practice on Java Island at 44.58% than on non-Java Islands at 41.62% (*p*-value 0.00001). According to this finding, some factors, such as better healthcare awareness, superior economic strength, and advanced healthcare facilities on Java Island, have potent contributions [68]. Interventions to strengthen BSE practice should include an exclusive focus on the non-Java Islands population, where subnational regions tended to practice inadequately. We suggested that cultural and social diversities should accompany health, financial, and technical support to increase BSE practice. Innovative programs in healthcare services should be explored, and investing in high-quality education, infrastructure, and communication systems must be boosted [69].

This meta-analysis revealed that women with lower educational status were less likely to perform BSE than women with higher educational status. The analysis of BSE practice among four groups based on the study population and settings revealed a higher tendency to perform BSE in the university students’ group at 49.90%, followed by school-aged students (42.51%), fertile women in Public Health Care/PHC (40.66%), and fertile women in communities (39.06%). The knowledge exposure of BSE practice was more likely responded to by women with university educations and supported by appropriate and higher health educational information within the environment. This finding also is supported by a study conducted in Ethiopia by Yeshitila et al. [9]. However, the lowest percentage of BSE practice was noted in fertile women in the communities group and should be given further attention. Approaches to improve breast education should be consolidated and expanded into the community sector or even individuals. A particular account must be sighted on the poor, less educated, and rural populations with a specific approach adapted according to local behaviors [69].

Other studies from Iran [70,71], Ghana [72,73], Nigeria [74], and Vietnam [11] supported our findings. Lower educational status is associated with a lower tendency of women to explore and receive breast health information. Furthermore, their intention toward personal healthcare and the opportunity to gain personal experience may be hampered. Raghupathi et al. stated the development of various skills and traits (including learned effectiveness, individual control, cognitive, and problem-solving abilities) due to educational exposure predisposes people towards improved health outcomes [75]. Therefore, we advised that more BC screening awareness and BSE practice exposure should be focused on women with low educational levels [76]. Our finding on breast cancer information exposure also supported the importance of breast cancer information that could shift the women’s behavior to perform BSE.

We also noticed a new phenomenon in several studies specifically conducted on young women mentioned that knowledge sharing via social media (such as WhatsApp, Instagram, Youtube, etc.) also encourages and helps young women learn BSE practices, especially in the COVID-19 pandemic era [39,77]. We believe that knowledge sharing via social media should be used more by the Indonesian government to enhance BSE practice in young women’s communities. This sharing via social media could also be a useful tool to spread knowledge of BSE to rural communities in remote areas in Indonesia. For those regions without internet access, we also suggest knowledge-sharing enhancements by family physicians or public health workers in the local community [78,79]. 

We noticed that family history of BC and family support drove women to perform BSEs. Family values and opinions are still closely tied to the cultural beliefs of Asian cultures and affect healthcare awareness [21,25,26,27,28,35,46]. This social phenomenon has proven true in Indonesian society, as shown in this study. These results reflected women’s willingness to check themselves when the family supports them to do so and if they do not consider it as a taboo or weird thing to do. Some of the participants also stated fear about the family history of breast cancer that might happen to them, thus leading them to practice BSE regularly. We conclude that their apprehension and surrounding support from family led to positive feedback on themselves, affecting the BSE participants. Studies from Turkey [80], Malaysia [81], and Saudi Arabia [82] supported these findings. This is also an important note for healthcare providers, as to enhance BSE practice, they need to educate the whole family more about the necessity of its practice in finding a concerning breast mass before it’s too late.

In this study, we also found an important additional finding (and we also acknowledge this as our limitation): most of the included studies (40 of 41 studies) were conducted in the western region of Indonesia. Only 1 study of BSE practices prevalence from Papua Island (the easternmost region of our country). Studies from other Indonesian east regions, such as Nusa Tenggara Timur and Maluku, were not found. It is known that the western region of Indonesia is relatively more economically and socially prosperous than the eastern region of Indonesia [12]. The eastern region of Indonesia has difficulty in healthcare deployment due to its demanding geographical nature, and much of its population is still living below Indonesian social welfare standards [12,14]. Therefore, we also suggest that studies of BSE practice prevalence should be conducted in these regions to give better BC screening and care for women in those areas.

We also noticed several other limitations in our study. All of the included studies are only cross-sectional studies; therefore, there could be recall bias. Additionally, all of the studies did not mention the quality, frequency, or regularity of BSE practice. An unequal proportional ratio of 1:3 total samples and most studies on non-Java Islands derived from western regions of Indonesia cause an unbalanced comparison between Java and non-Java Islands; thus, the findings may not be representative. Additionally, as we mentioned before, Indonesia consists of a highly diverse population; therefore, this study could not represent overall Indonesian society. We also admit that the high heterogeneity of the Indonesian population cannot be explained using meta-regression performed in the included studies, which influence the statistical calculations and their significance.

## 5. Conclusions

The overall national prevalence of BSE practice in Indonesia was significantly low (less than 50%), particularly in non-Java Islands. Higher education, knowledge of BC, attitude toward BSE, family history of BC, family support on BSE, and BC information exposure influenced participation in BSE practice. Collaborative programs should be enhanced with the government, health policymakers, and community health groups at the individual or community level to promote BSE as routine screening. Even though these findings are statistically significant, more well-structured, in-depth epidemiological analyses are needed to address unanswered questions, especially in Indonesia’s eastern regions.

## Figures and Tables

**Figure 1 diagnostics-13-02577-f001:**
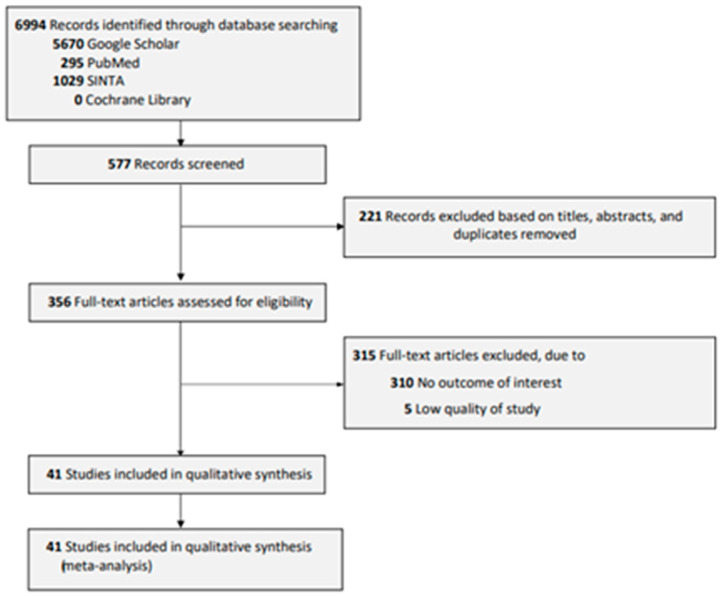
Flow Chart of Article Selection.

**Figure 2 diagnostics-13-02577-f002:**
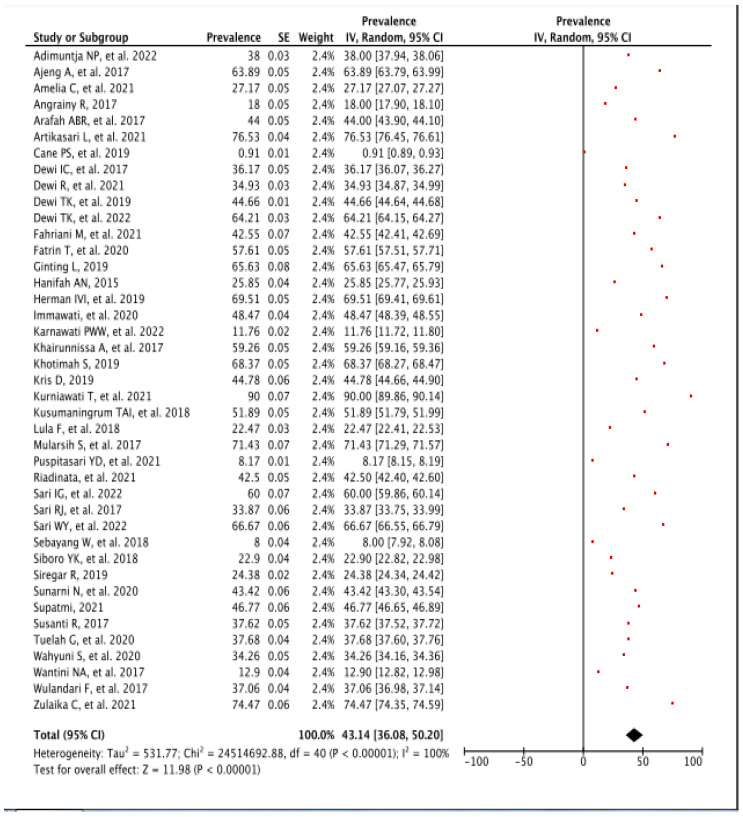
Forest Plot of Overall BSE Prevalence. CI—Confidence Interval; df—Degree of Freedom; *I*^2^—I squared; IV—Inverse Variance; SE—Standard Error.

**Figure 3 diagnostics-13-02577-f003:**
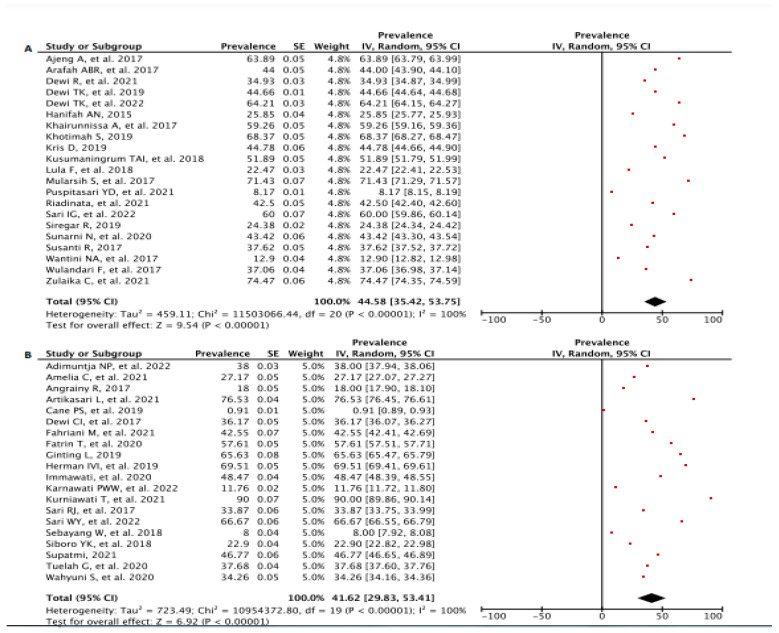
Forest Plot of BSE Prevalence in (**A**) Java vs. (**B**) non-Java Region. CI—Confidence Interval; df—Degree of Freedom; *I*^2^—I squared; IV—Inverse Variance; SE—Standard Error.

**Figure 4 diagnostics-13-02577-f004:**
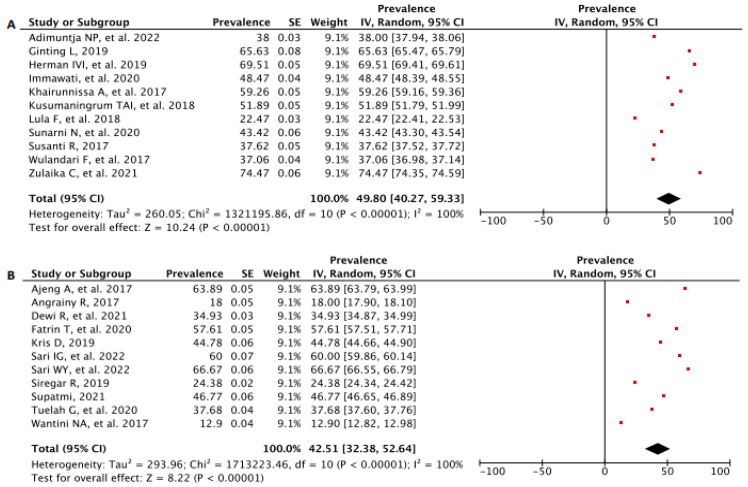
Forest Plot of BSE prevalence analysis of BSE practice based on the study population: (**A**) university students, (**B**) school-aged students, (**C**) fertile women and public health center-based (PHC), and (**D**) fertile women in communities. CI—Confidence Interval; df—Degree of Freedom; *I*^2^—I squared; IV—Inverse Variance; SE—Standard Error.

**Figure 5 diagnostics-13-02577-f005:**
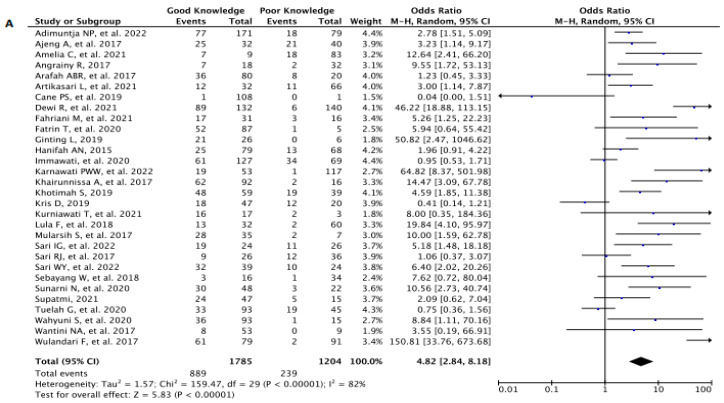
Forest Plot to analyze the correlation between (**A**) knowledge, (**B**) attitude, and (**C**) family history of BC, each to BSE Prevalence. CI—Confidence Interval; df—Degree of Freedom; *I*^2^—I squared; IV—Inverse Variance; SE—Standard Error.

**Figure 6 diagnostics-13-02577-f006:**
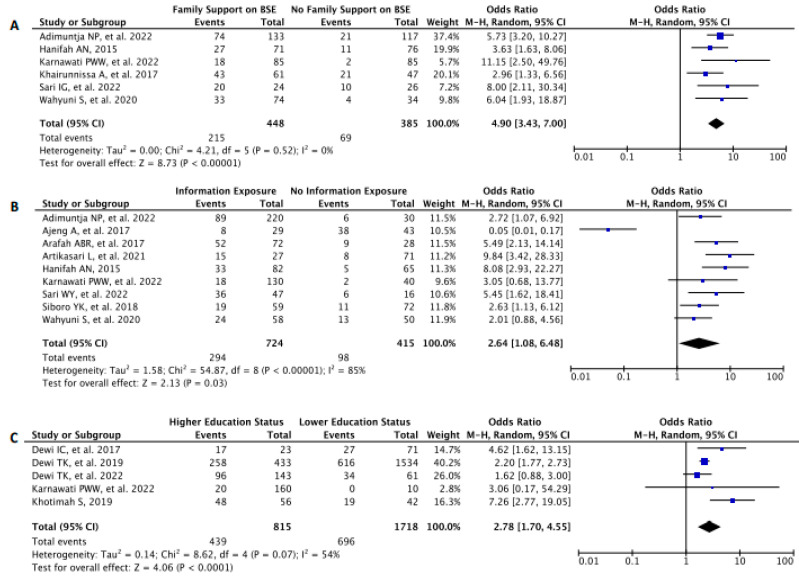
Forest Plot to analyze correlation between (**A**) family support, (**B**) information exposure, and (**C**) education status, each to BSE prevalence. CI—Confidence Interval; df—Degree of Freedom; *I*^2^—I squared; IV—Inverse Variance; SE—Standard Error.

**Figure 7 diagnostics-13-02577-f007:**
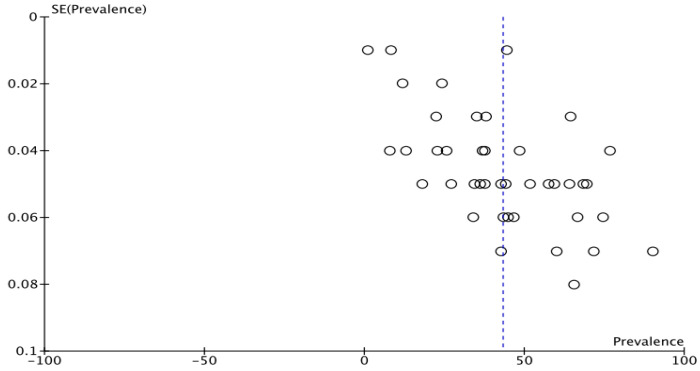
The national prevalence funnel plot. SE—Standard Error.

**Table 1 diagnostics-13-02577-t001:** Summarized data characteristics of the included studies of breast self-examination practice and its determinants among women in Indonesia.

Authors, Year	Study Area, Region	Study Setting	Study Design	Study Population	Age	Sample Size
Adimuntja NP et al. 2022 [21]	Jayapura, Non-Java	University-based	Cross-sectional	Students	18–23	250
Ajeng A et al. 2017 [22]	Tangerang, Java	School-based	Cross-sectional	Students	NI	72
Amelia C et al. 2021 [23]	Batam, Non-Java	PHC-based	Cross-sectional	Fertile women	NI	92
Angrainy R, 2017 [24]	Pekanbaru, Non-Java	School-based	Cross-sectional	Students	15–17	50
Arafah ABR et al. 2017 [25]	Surabaya, Java	Community-based	Cross-sectional	Fertile women	40–50	100
Artikasari L et al. 2021 [26]	Jambi, Non-Java	PHC-based	Cross-sectional	Fertile women	NI	98
Cane PS et al. 2019 [27]	Aceh, Non-Java	Community-based	Cross-sectional	Fertile women	20–45	110
Dewi IC et al. 2017 [28]	Denpasar, Non-Java	PHC-based	Cross-sectional	Fertile women	20–49	94
Dewi R et al. 2021 [29]	Sukabumi, Java	School-based	Cross-sectional	Students	15–18	272
Dewi TK et al. 2019 [30]	Surabaya, Java	Community-based	Cross-sectional	Fertile women	20–60	1967
Dewi TK et al. 2022 [31]	Surabaya, Java	Community-based	Cross-sectional	Fertile women	18–61	204
Fahriani M et al. 2021 [32]	Bengkulu, Non-Java	PHC-based	Cross-sectional	Fertile women	NI	47
Fatrin T et al. 2020 [33]	Palembang, Non-Java	School-based	Cross-sectional	Students	NI	92
Ginting L, 2019 [34]	Medan, Non-Java	University-based	Cross-sectional	Students	17–25	32
Hanifah AN, 2015 [35]	Surakarta, Java	PHC-based	Cross-sectional	Fertile women	15–49	147
Herman IVI et al. 2019 [36]	Kupang, Non-Java	University-based	Cross-sectional	Students	16–24	82
Immawati et al. 2020 [37]	Lampung, Non-Java	University-based	Cross-sectional	Students	17–21	196
Karnawati PWW et al. 2022 [38]	Mataram, Non-Java	Community-based	Cross-sectional	Fertile women	20–49	170
Khairunnissa A et al. 2017 [39]	Jakarta, Java	University-based	Cross-sectional	Students	NI	108
Khotimah S, 2019 [40]	Tangerang, Java	PHC-based	Cross-sectional	Fertile women	15–50	98
Kris D, 2019 [41]	Kediri, Java	School-based	Cross-sectional	Students	NI	67
Kurniawati T et al. 2021 [42]	Lampung, Non-Java	Community-based	Cross-sectional	Fertile women	15–21	20
Kusumaningrum T et al. 2018 [43]	Surakarta, Java	University-based	Cross-sectional	Students	20–23	106
Lula F et al. 2018 [44]	Jember, Java	University-based	Cross-sectional	Students	≥20	227
Mularsih S et al. 2017 [45]	Semarang, Java	Community-based	Cross-sectional	Fertile women	15–40	42
Puspitasari YD et al. 2021 [46]	Jember, Java	Community-based	Cross-sectional	Teenagers	10–19	60
Riadinata et al. 2021 [47]	Bantul, Java	Community-based	Cross-sectional	Fertile women	20–35	91
Sari IG et al. 2022 [48]	Bogor, Java	School-based	Cross-sectional	Students	NI	50
Sari RJ et al. 2017 [49]	Medan, Non-Java	Community-based	Cross-sectional	Fertile women	20–49	62
Sari WY et al. 2022 [50]	Palembang, Non-Java	School-based	Cross-sectional	Students	NI	63
Sebayang W et al. 2018 [51]	Medan, Non-Java	PHC-based	Cross-sectional	Fertile women	25–49	50
Siboro YK et al. 2018 [52]	Pekanbaru, Non-Java	Community-based	Cross-sectional	Fertile women	15–49	131
Siregar R, 2019 [53]	Karawang, Java	School-based	Cross-sectional	Students	NI	320
Sunarni N et al. 2020 [54]	Ciamis, Java	University-based	Cross-sectional	Students	NI	76
Supatmi, 2021 [55]	Palembang, Non-Java	School-based	Cross-sectional	Students	NI	62
Susanti R, 2017 [56]	Jakarta, Java	University-based	Cross-sectional	Students	NI	118
Tuelah G et al. 2020 [57]	Manado, Non-Java	School-based	Cross-sectional	Students	NI	138
Wahyuni S et al. 2020 [58]	Palangkaraya, Non-Java	Community-based	Cross-sectional	Teenagers	NI	108
Wantini NA et al. 2017 [59]	Sleman, Java	School-based	Cross-sectional	Students	NI	62
Wulandari F et al. 2017 [60]	Kuningan, Java	University-based	Cross-sectional	Students	20–30	170
Zulaika C et al. 2021 [61]	Semarang, Java	University-based	Cross-sectional	Students	17–21	47

NI—No Information; PHC—Public Health Center.

## Data Availability

Data supporting reported results can be found at: https://doi.org/10.5281/zenodo.7983569, accessed on 30 May 2023.

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
