# Peer review of "Breast Self-Examination Practice and Its Determinants among Women in Indonesia: A Systematic Review, Meta-Analysis, and Meta-Regression"

_diagnostics, 2023, doi:10.3390/diagnostics13152577_

Round 1

Reviewer 1 Report

1. Table 1 is not prepared and especially not organized at the end.

2. Figures 2 to 6 are presented in different letter size formats, and nothing can be understood from them.

3. There is nothing new in the presented data, the presented data that SBE is not associated with other diagnostic and mandatory tests, such as breast US, MRI, and biopsy are performed only in large medical centers and depend on women's education, also do not say anything. These are more general demographic data, from which systematic conclusions and decisions for the future should be made.

No significant comments

Author Response

Reviewer 1

We thank you for your gracious review and comments on our article. We would like to address several issues that you raised:

  1. Does the introduction provide sufficient background and include all relevant references? Your opinion: Yes
  2. Are all the cited references relevant to the research: Yes
  3. Is the research design appropriate: it could be improved.

Response: We realized that all the included studies were only cross-sectional; none was cohort or case-control. We acknowledged this, and we added in the study limitations. ( page 13, last paragraph of discussion)

  1. Are the methods adequately described: yes
  2. Are the result presented: must be improved.

Response: We revised Table 1 and enhanced all font sizes in all figures. We hope it is now more presentable. We also add some comments before the table and figures, and we hope they clarify the result.

  1. Are the conclusions supported by the result: can they be improved. We hope after we revise the result section, this is no longer a problem.

Your individual comments :

  1. Table 1 is not prepared and especially not organized at the end.

Response: We have already revised this problem.

  1. Figures 2-6 are presented in different letter size formats, and nothing can be understood.

Response: We already revised this problem; we enhanced all font sizes in all figures and hope it is now more presentable. We also add some comments before the table and figures. We hope it adds clarity to understanding the result.

  1. There is nothing new in the presented data; the data in BSE is not associated with other diagnostic and mandatory tests, such as breast US, MRI and biopsy are performed only in large medical centers and depend on women’s education. Also, do not say anything. These are more general demographic data from which systematic conclusions and decisions for the future should be made.

Response :

In the last paragraph's introduction, we clearly stated that this article's purpose is to synthesize the pooled prevalence data of BSE practice, compare BSE practice prevalence between Java Island and non-Java Islands in Indonesia and identify the determinants which we thought could affect the BSE practice in Indonesian population. Therefore, it is clear that in this article, we didn’t associate BSE with other diagnostic tools or biopsies.

We also acknowledged that BSE is not a standard early detection tool for Breast Cancer. But in Indonesia, it is used widely as an early detection method for Breast cancer, and even it is encouraged in Indonesia (due to many problems we faced and mentioned in the article).

However, we truly appreciate your comment, and we comprehend that the same opinion as your comment could be raised in the readers’ minds later. Therefore, we add that BSE only detects a palpable mass. ( introduction, 4th paragraph).

Reviewer 2 Report

The review from Azhar et al. is an excellent insight into the problematics of BSE in Indonesia. I recommend accepting the article after minor revisions. 

1. Introduction: I recommend shortening the paragraphs about healthcare in Indonesia to essential information only. 

2. Methods: 115 -  "Regular BSE is a routine once-a-month examination toward the end of the menstrual period." This sentence should be listed in Introduction, not in Data items. 

3. Table 1: The results are mixed together, better table formatting should be done.  

4. Discussion - 311 - Please also compare the prevalence of BSE only in Europe, not only mammography screening. 

5. Discussion - I encourage the authors to discuss also the quality of performed BSE, because we don't know if the women in mentioned studies performed the BSE correctly. 

6. - 357 - "Harvey et al" It seems there should be a citation in the form of a number. 

7. 381-384, 403 - redundant spaces before and after the brackets

Author Response

Reviewer 2

We thank you for your gracious review and comments on our article. We would like to address several issues that you raised :

Your comments :

  1. Introduction: I recommend shortening the paragraphs about healthcare in Indonesia to essential information only. 

Response: We have shortened the paragraphs about healthcare in Indonesia.

  1. Methods: 115 -  "Regular BSE is a routine once-a-month examination toward the end of the menstrual period." This sentence should be listed in Introduction, not in the Data items.

Response: We have deleted the sentence in the Data Item.

  1. Table 1: The results are mixed, better table formatting should be done.  

Response: We have resolved the formatting issue.

  1. Discussion - 311 - Please also compare the prevalence of BSE only in Europe, not only mammography screening. 

Response: We only found one small study for BSE prevalence in Europe; we already add this in the 3rdparagraph of the discussion. We decided to erase the paragraph about mammography in Europe because it did not align with the discussion topic.

  1. Discussion: I encourage the authors to discuss also the quality of performed BSE because we don’t know if the women in mentioned studies performed the BSE correctly.

Response: We didn’t find it in all the studies. Therefore we add it to the study limitations  (discussion, last paragraph).

  1. 357 - "Harvey et al." It seems there should be a citation in the form of a number. 

Response: We have resolved the issue.

  1. 381-384, 403 - redundant spaces before and after the brackets

Response: We have resolved the issue.

Round 2

Reviewer 1 Report

no serious comments

no serious comments